:ᐧᗺᒧᗡᓯ PLOS | ONE

# MARGO (Massively Automated Real-time GUI for Object-tracking), a platform for high-throughput ethology

Zach Werkhoven[1], Christian Rohrsen[1,2], Chuan Qin[1], Björn Brembs[2], Benjamin de Bivort[1]*

**1** Dept. of Organismic and Evolutionary Biology & Center for Brain Science, Harvard University, Cambridge, MA, United States of America, **2** Institut für Zoologie - Neurogenetik, Universität Regensburg, Regensburg, Germany

* debivort@oeb.harvard.edu

**Data Availability Statement:** MARGO's code is available in the MARGO repository on github (https://github.com/de-Bivort-Lab/margo). All behavioral data is available on Zenodo

## Abstract

Fast object tracking in real time allows convenient tracking of very large numbers of animals and closed-loop experiments that control stimuli for many animals in parallel. We developed MARGO, a MATLAB-based, real-time animal tracking suite for custom behavioral experiments. We demonstrated that MARGO can rapidly and accurately track large numbers of animals in parallel over very long timescales, typically when spatially separated such as in multiwell plates. We incorporated control of peripheral hardware, and implemented a flexible software architecture for defining new experimental routines. These features enable closed-loop delivery of stimuli to many individuals simultaneously. We highlight MARGO's ability to coordinate tracking and hardware control with two custom behavioral assays (measuring phototaxis and optomotor response) and one optogenetic operant conditioning assay. There are currently several open source animal trackers. MARGO's strengths are 1) fast and accurate tracking, 2) high throughput, 3) an accessible interface and data output and 4) real-time closed-loop hardware control for for sensory and optogenetic stimuli, all of which are optimized for large-scale experiments.

## Introduction

Automated animal tracking methods have become commonplace in the study of behavior. They enable large sample sizes, high statistical power, and more rapid inference of mechanisms giving rise to behavior. Existing animal trackers vary in computational complexity and are often specialized for particular imaging configurations or behavioral measurements. Trackers can assist in a wide range of experimental tasks such as monitoring activity, measuring response to stimuli [1, 2], and locating body parts over time [3, 4]. Some trackers are designed to track and maintain identities of multiple individuals occupying the same arena [5–8] while others measure the collective activity of groups without maintaining identities or rely on physical segregation of animals to ensure trajectories never collide [9–12]. But few of these trackers

(https://zenodo.org/record/2596143#.XQP1-vlKiRd). Instrument schematics are available on github at: de Bivort Lab schematics repository (https://github.com/de-Bivort-Lab/dblab-schematics).

**Funding:** This research was supported by NSF Graduate Research Fellowship (DGE-1144152) to ZW; an iPUR Fellowship and a Fulbright Fellowship to CR; NSF (IOS-1557913), Alfred P. Sloan Foundation, Klingenstein-Simons Fellowship, and Smith Family Foundation to BLdB. The funders had no role in study design, data collection and analysis, decision to publish, or preparation of the manuscript.

**Competing interests:** BLdB is a scientific advisor of FlySorter, LLC. This does not alter our adherence to PLOS ONE policies on sharing data and materials.

are designed as platforms for high throughput, hardware control, and flexible experimental reconfiguration.

Improvements in machine learning and template matching approaches to object localization and classification have made it possible to efficiently train models that accurately track and classify a variety of animal species and visually distinguish identities of individuals across time [5, 6, 8, 13]. Tracking individual identity in groups requires resolving identities through collisions where bodies are overlapping. FlyTracker and idTracker.ai train classifiers to assign identities to individuals in each frame and also extract postural information such head and limb position. In optimized experiments, these trackers can maintain distinct identities over extended periods with minimal human intervention. Other trackers, such as Ctrax ToxTrac, and Tracktor [7, 14, 15], track animals by segmenting them from the image background and assign identities by stitching traces together across frames based on changes in position. Although the classification accuracy can be quite high under optimal conditions, these methods generally require human intervention to prevent assignment error from propagating over longer timescales even at low error rates (or they are used for analyses where individual identity is not needed).

Both approaches to identity tracking can be used to study complex social and individual behaviors, but the computational cost of collision resolution means that tracking is generally performed offline on recorded video data [16]. Furthermore, the need to record high-quality, high-resolution video data can make it challenging to track animals over long experiments. Some methods of postural segmentation require manual addition of limb markers [17], splines fit in post-processing [18], or computationally heavy machine vision in post-processing [3, 4, 8]. In all cases, the need to separate tracking and recording can be rate-limiting for experiments. Real-time tracking offers the benefits of allowing closed-loop stimulus delivery and a small data footprint due to video data not being retained. In general, real-time tracking methods are less capable of tracking individuals through collisions because they cannot use future information to help resolve ambiguities [11]. For that reason, real-time multiple animal trackers can fall back on spatial segregation of animals to distinguish identities or dispense with identity tracking altogether [12, 16]. Some existing real-time trackers can track multiple animals (without maintaining their identity through collisions) in parallel and support a variety of features such as modular arena design, and closed-loop stimulus delivery [19–22].

The tracking algorithms, software interface, hardware configurations, and experimental goals of available trackers vary greatly. Some packages such as Tracktor and FlyWorld use a simple application programming interface (API) and implement tracking through background segmentation and match identities with Hungarian-like Algorithms that minimize frame-to-frame changes in position [7, 16, 23]. Ethoscopes are an integrated hardware and software solution that take advantage of the small size and low cost of microcomputers such as the Raspberry Pi. They support modular arenas and peripheral hardware for stimulus delivery [19] and can be networked and operated through a web-based interface to conduct experiments remotely and at scale. Ethoscopes provide a hardware template and API for integrating peripheral components into behavioral experiments, but the Ethoscope tracker is not currently designed to operate independent of the hardware module. BioTracker offers a graphical user-interface (GUI) that allows the user to select from different tracking algorithms with easily customized tracking parameters or import and use a custom algorithm [24].

We wanted a platform that integrated many of the positive features of these trackers into a single software package, while supporting genome-scale screening experiments in a flexible way that would support the needs of labs that study diverse behaviors. We prioritized 1) fast and accurate individual tracking that could be scaled to very large numbers of individuals or experimental groups over very long timescales, 2) flexibility in the user interface that would

permit a diversity of organisms, tracking modes, experimental paradigms, and behavioral arenas, 3) integration of peripheral hardware to enable closed-loop sensory and optogenetic stimuli, and 4) a user-friendly interface and data output format.

We developed MARGO, a MATLAB based tracking suite, with these goals in mind. MARGO can reliably track up to thousands of individuals simultaneously in real-time for days or longer (with limits only set by logistical challenges such as keeping animals fed). MARGO has two tracking modes that allow it to distinguish either individuals or groups of individuals that are spatially segregated. We show that traces acquired in MARGO are comparably accurate to those of other trackers and are robust to noisy images and changing imaging conditions. We also demonstrate that tracking works reliably with nonspecialist equipment (like smart phone cameras). MARGO provides visual feedback on tracking performance that streamlines parameter configuration, making it easy to setup new experiments.

Additionally, MARGO can control peripheral hardware, enabling closed-loop individual stimulus delivery in high-throughput paradigms. Using adult fruit flies, we demonstrate three closed-loop [25] applications in MARGO for delivering individualized stimuli to multiple animals in parallel. First we measured individual phototactic bias in Y-shaped arenas. Second we quantified individual optomotor response in circular arenas. In the third assay, we configured MARGO to deliver optogenetic stimulation in real-time. Though MARGO was developed and tested with adult fruit flies, we show that it can be used to track many organisms such as fruit fly larvae, nematodes, larval zebrafish and bumblebees. We packaged MARGO with an easy-to-use graphical user interface (GUI) and comprehensive documentation to improve the accessibility of the software and offer it as a resource to the ethology community. Though it does not perform visual identity recognition or postural limb tracking, we believe that MARGO can meet the needs of many large behavioral screens, experiments requiring real-time stimulus delivery, and users looking to run rapid pilot experiments with little setup.

## MARGO workflow

The core experimental workflow of a MARGO experiment (Fig 1A) can be briefly summarized as follows: 1) define spatial regions of interest (ROIs) in which flies will be tracked, 2) construct a background image used to separate foreground and background, 3) compute statistics on the distribution of the number of foreground pixels under clean tracking conditions to facilitate detection and correction of noisy imaging, 4) perform tracking. We found that constraining the space in which an animal might be located significantly relaxed the computational requirements of multi-animal tracking. Because MARGO is designed for high-throughput experiments, it needs to be convenient to define up to thousands of ROIs. MARGO has two modes for defining ROIs. The first is automated detection that detects and segments regular patterns of high-contrast regions in the image, such as back-lit arenas. The second prompts the user to manually place grids of ROIs of arbitrary size. In practice, we find that ROI definition typically takes a few seconds but can take as long a few minutes.

Following ROI definition, a background image is constructed for each ROI separately. Each background image is computed as the mean or median (as configured by the user) image from a rolling stack of background sample images. Tracking is performed by segmenting binary blobs from a thresholded difference image computed by subtracting each frame from an estimate of the background (Fig 1B). Background subtraction commonly suffers from two issues with opposing solutions. The first is that subtle changes in the background over time introduce error in the difference image, requiring continuous averaging or reacquisition of the background image. The second is that continuous averaging or reacquisition of the background can make inactive animals appear as part of the background rather than foreground, making

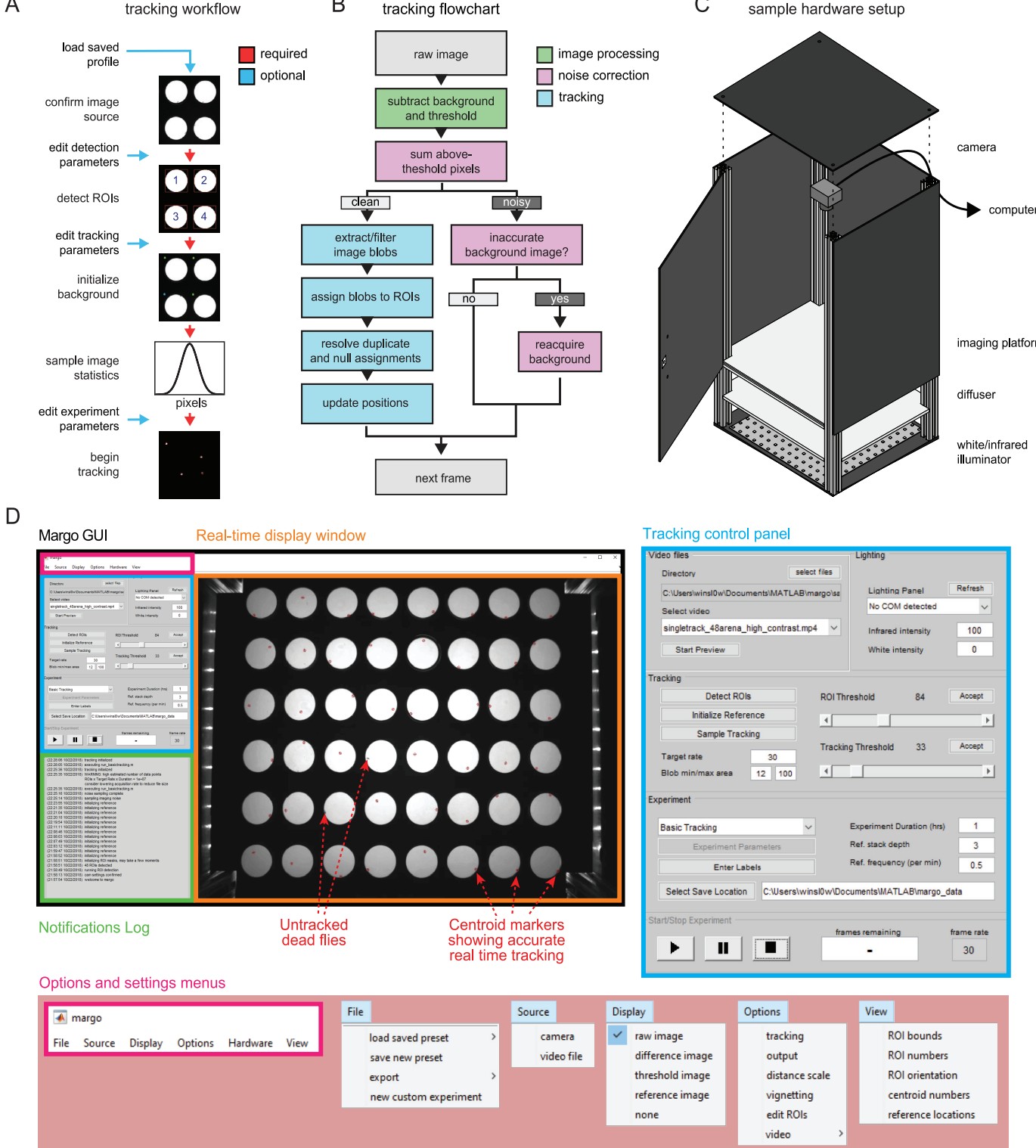

**Fig 1. MARGO workflow, tracking algorithm, and sample behavioral box.** A) Diagram of the user workflow to set up a new tracking experiment. Arrow color indicates whether the setup step is required. Before tracking, users define an input source, define ROIs to track, initialize a background image used to separate foreground and background, and sample the image statistics on a reference of clean tracking. Tracking parameters can be customized at multiple points (blue arrows). B) Flowchart depicting the MARGO's frame-to-frame tracking routine. Each frame consists of image processing (green) to segment foreground from the background, noise estimation (magenta) to assess the quality of foreground segmentation and determine if the current frame can be tracked, and tracking (cyan) of

foreground binary blobs. MARGO's tracking algorithm skips noisy frames and re-acquires the background image if many consecutive frames are deemed too noisy to track. C) Schematic of a typical behavioral box used for tracking. Behavioral arenas are backlit with an LED illuminator and imaged with an overhead camera. The tracking camera is fitted with an infrared filter to allow light visible to the animals to be controlled independently of the tracking illumination. A diffuser panel between the LED backlight and the behavioral arenas makes the illumination even. The camera and illuminator are both connected to a computer for real-time tracking and control via MARGO. D) Representative views of MARGO's GUI. Blue inset shows the controls for setting tracking parameters, pink inset the menu options for configuring experiments.

them undetectable in the thresholded difference image. Constructing the reference for each ROI separately mitigates these concerns by allowing the reference to be constructed in a piecemeal fashion by adding a background sample image only when the animals have moved from the positions they occupied in previous images of the background stack. The time needed to establish a background image depends on the activity level of the animals and the number of images in the reference stack. We typically find that 3-30 seconds are needed to initialize the background image. Once a background image is established, tracking can begin. In each frame, candidate blobs are identified as the blobs that are both 1) between minimum and maximum size threshold and 2) located within the bounds of an ROI. Candidate blobs are subsequently assigned to ROIs by spatial location. Within each ROI, candidate blobs are matched to centroid traces by minimizing the total frame-to-frame changes in position within each ROI. If the number of candidates exceeds the number of traces in a given ROI, only the candidates closest to the last centroid positions of the traces are assigned. If the number of traces exceeds the number of candidates, the candidates are assigned to the closest traces and any remaining traces are assigned no position (i.e., NaN for that frame).

Degradation of difference image quality over time (due changes in the background, noisy imaging, and physical perturbation of the imaging setup) constitutes a significant barrier to long term tracking [15]. To address this problem, MARGO continuously monitors the quality of the difference image and updates or reacquires the background image when imaging becomes noisy. We refer to this collective process as noise correction. Prior to tracking, MARGO samples the distribution of the total number of above-threshold pixels under clean imaging conditions to serve as a baseline for comparison. During tracking, the software then continuously calculates that distribution on a rolling basis and reacquires a background image when the rolling sample substantially deviates from the baseline distribution.

## Tracking accuracy and noise robustness

We performed a number of experiments and analyses to assess MARGO's robustness to tracking errors and comparability with other trackers. In these experiments, we tracked individual flies, each alone in a circular arena, so that individual identity was assured by spatial segregation.

We assessed the ability of MARGO to handle degradation of the difference image by repeatedly shifting the background image by a small amount in a random direction (2px, 2% of the arena diameter, and 0.16% of the width of the image) to mimic situations where an accidental nudge or vibration shifts the arena. MARGO was used to simultaneously record a movie of individual flies walking in circular arenas and track their centroids. These tracks were the ground truth for this misalignment experiment, and background shifting was implemented digitally on the recorded movie. MARGO reliably detects the changes in difference image statistics associated with each of these events and recovers clean tracking by reacquiring the background, typically within 1 second (Fig 2A and 2B). Forcing reacquisition of the background image has the disadvantage of resetting the reference with a single image, meaning that a normal background image built by median-filtering multiple frames spaced in time cannot be computed immediately (background images made this way have two benefits: lower pixel

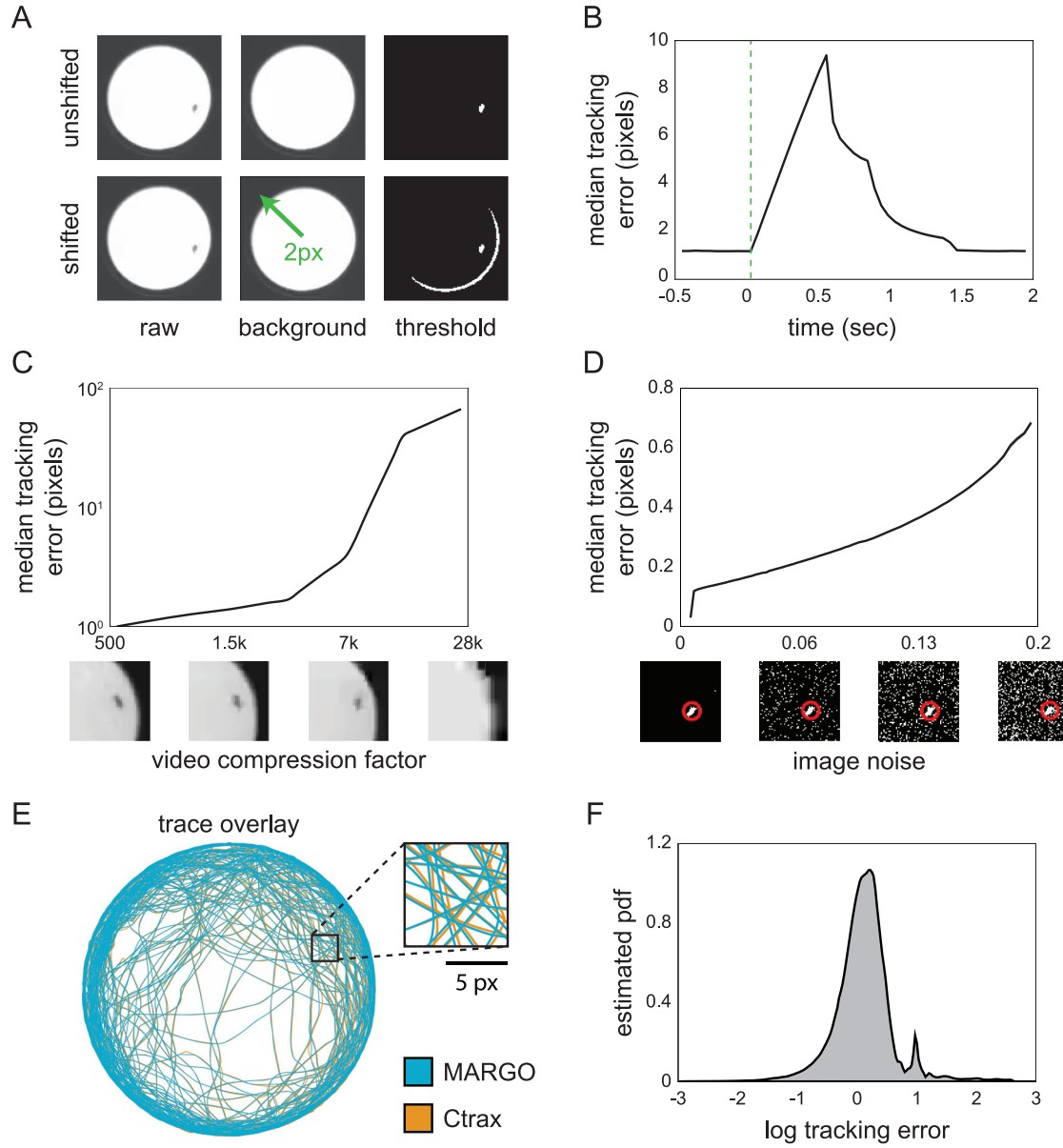

**Fig 2. MARGO tracking accuracy and robustness to imaging noise.** A) Diagram of the background image shifting scheme used to simulate the kind of background inaccuracy that can happen in long experiments. B) Trial-triggered median tracking error centered on reference shifting. C) Median error of tracking performed on the same video at different levels of compression. Below: sample images. D) Median tracking error versus different levels of added noise. Pixel noise was manually added to the binary threshold image downstream. Below: sample images with estimated fly position (red circle). E) Sample trace comparison and F) log distribution of tracking error between traces acquired from the same video in both MARGO and Ctrax. The 95% confidence interval of the above means are shown but are within the line thickness.

noise and fewer tracking dead spots because they do not include moving animals). This typically caused a reduction in tracking accuracy that is brief (<2s) and had little effect on the overall correlation of the tracking data to the ground truth (r = 0.9998). Indeed, we found a small effect on tracking error (mean 3.07± 2.5 pixels, which corresponds to 20% of a fly's body length at our typical imaging resolution) even when shifting the background every 2 seconds. In our experimental set-ups, noise-induced background reacquisition was relatively rare, typically occurring fewer than 10 times over the duration of a two hour experiment.

We tested MARGO's sensitivity to video compression by compressing and tracking a video previously captured during a real-time tracking session. The centroid position error of traces acquired from compressed videos were calculated by comparing them to the ground-truth traces acquired on uncompressed images in real time. MARGO showed sub-pixel median tracking error up to 3000-fold compression (Fig 2C). We further tested the robustness of MARGO to noisy imaging by digitally injecting pixel noise (by randomly setting each pixel to True with a fixed probability) into the thresholded difference image of each frame of a video previously acquired and tracked under clean conditions. Noise was added downstream of noise correction and upstream of tracking to simulate tracking under conditions where noise correction is poorly calibrated. We observed sub-pixel median tracking error up to 20%pixel noise (Fig 2D). In practice, we find it easy to create imaging conditions with noise levels <1% pixel noise without the use of expensive hardware.

To compare the tracking accuracy of MARGO to a widely used animal tracker, we fed uncompressed video captured during a live tracking session in MARGO into Ctrax [14] and measured the discrepancy between the two sets of tracks. Overall we found a high degree of agreement between traces acquired in MARGO and Ctrax (Fig 2E and 2F). We attribute the majority of discrepancies to minor variations in blob size and shape arising from differences in background segmentation. It is worth noting that although Ctrax flagged many frames for manual inspection and resolution, for comparability we opted not to resolve these frames and instead restricted our analysis to the automatically acquired traces. (Ctrax primarily uses these flags to draw user attention to tracking ambiguities through collisions, which did not happen in our experiment because flies were spatially segregated.) Manual inspection of tracked frames with error larger than 1 pixel revealed that most major discrepancies occurred in one of two ways: 1) short periods between the death and birth of two traces on the same animal in Ctrax, or 2) identity swaps in Ctrax between animals in neighboring arenas. These errors may be attributable to our inartful use of Ctrax.

## High-throughput behavioral screens

We designed MARGO with high-throughput behavioral screens in mind, with hundreds of experimental groups, each potentially containing hundreds or thousands of animals. Many features in MARGO's GUI have been included to reduce the time needed to establish successful tracking, including automated ROI detection and visualizations of object statistics and the effects of parameters. Configuring tracking for experiments with hundreds of individuals typically took between 2-5 minutes. Additionally, we added the ability to save and load parameter and experimental configurations.

The speed of the tracking algorithm permits the tracking of very large numbers of animals simultaneously in a single field of view (facilitating certain experimental designs, like testing multiple experimental groups simultaneously). To demonstrate MARGO's throughput, we continuously tracked 960 flies at 8Hz for more than 6 days (S1 Video). Flies were singly housed in bottomless 96-well plates (Fig 3A) placed on top of food and were imaged by a single overhead camera. The appearance of the arenas changed substantially over 6 days due to evaporation of water from the fly food media, condensation on the well plate lids, and egg laying. Despite these changes, the quality of centroid traces and acquisition rates appeared stable throughout the experiment (Fig 3B). The overall activity level of flies decreased over the duration of the experiment (Fig 3C). The flies' log-speed distributions generally exhibited two distinct modes: a low mode consistent with frame-to-frame tracking noise and a higher mode consistent with movement of the flies (Fig 3D) [26, 27]. Individual flies varied in the relative abundance of these two modes. We defined a movement threshold as the local minimum

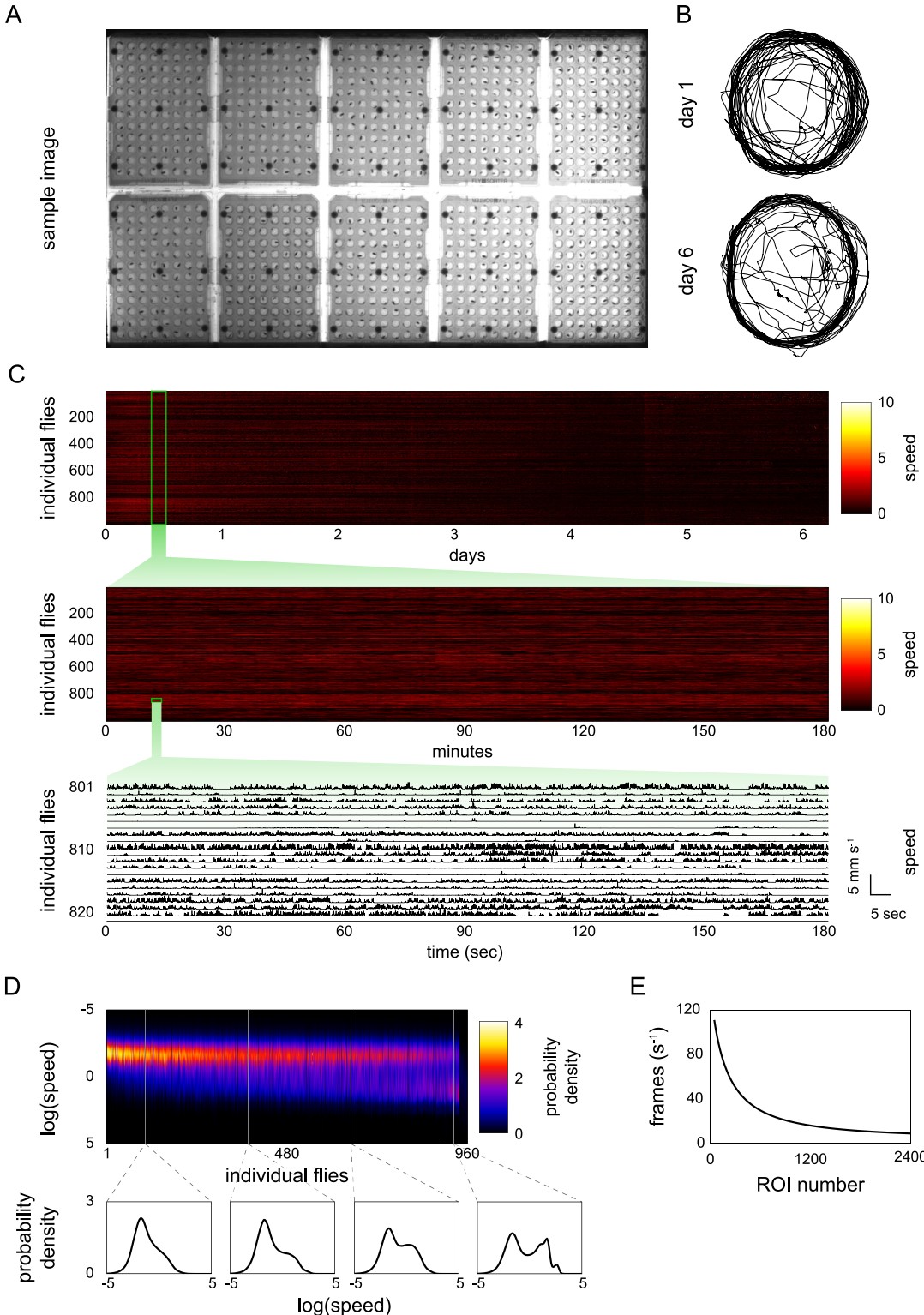

**Fig 3. MARGO tracking throughput.** A) Image of 10 single-fly housing plates from the overhead tracking camera. B) Sample tracks from the same fly on days 1 and 6. C) Fly speeds at three representative scales: heatmap of individual speed over the duration of the experiment (top), heatmap of individual speed from a three hour period (middle), raw speed traces from twenty individuals from a three minute period (bottom). Activity of most flies decreased over the six day duration. D) Individual kernel density estimates of log speed over the duration of the experiment. Column order was sorted by mean individual bout

length in ascending order. E) Acquisition frame rate as a function of number of ROIs tracked in a simulated experiment. The acquisition rate decreased exponentially, consistent with a linear increase in inter-frame interval as a function of ROI number.

between these two modes and parsed individual speed trajectories into movement bouts by identifying periods of continuous movement above the threshold. Sorting flies by the average length of their movement bouts revealed a trend of increasing mean and magnitude of the higher "movement" mode (Fig 3D), i.e., flies that walked longer tended to walk faster.

To measure MARGO's performance as a function of the number of ROIs, we recorded the mean real-time tracking rate while varying the number of tracked ROIs from a high-resolution (7.4MP) video composed of the same single-arena video repeated 2400 times in a grid. We found that the frame-to-frame latency scaled linearly as a function of the number of ROIs tracked (Fig 3E). On modern computer hardware (intel i7 4.0GHz CPU), we measured tracking rates of 160Hz for a single ROI down to 5Hz for 2400 ROIs. MARGO could plausibly track up to 5000 animals at lower rates (1 Hz), potentially fast enough for experiments monitoring changes in activity level changes over long timescales, like circadian experiments.

Large behavioral screens can potentially generate hundreds of hours of data on thousands of animals and massive data files even without recording videos. We found that experiments tracking many hundreds of animals over multiple days made raw data files too large to hold in memory on typical computers. We designed a custom data container and an API to easily work with data stored in large binary files. MARGO's raw data API includes methods to batch-process multiple tracking experiments or single datasets too large to hold in memory (see user documentation).

## Customization and versatility

To demonstrate MARGO's ability to prototype experiments without the need for specialized hardware, we ran a minimal tracking experiment using only commonly available materials. Individual fruit flies were placed into the wells of a standard 48 well culture plate. The plate was put in a cardboard box (to reduce reflections) on a sheet of white paper as a high contrast background. Movies were recorded on a 1.3MP smartphone camera using natural room light as illumination and imported into MARGO for tracking (S2 Video). Tracks and movement bouts acquired under these conditions showed no apparent differences to those acquired under our normal experimental conditions (custom arenas over diffused LED illuminators in light-sealed imaging boxes). However, we did find that the lower contrast illumination of this setup increased imaging noise and narrowed the range of parameters that worked for segmentation, but had no apparent effect on the accuracy of traces once calibrated.

MARGO was developed for high-throughput ethology in fruit flies, but many small organisms used for high-throughput behavior are more translucent than adult flies. To assess MARGO's tracking robustness on such organisms, we used MARGO to track videos of larval *Danio rerio*, *Caenorhabditis elegans*, larval *Drosophila* (S3–S5 Videos), and also bumblebees (*Bombus impatiens*) (S6 Video). As expected, the translucency of these organisms narrowed the functional range of some tracking parameters, but MARGO's real-time tracking feedback made it easy to dial in these parameters. Sample traces acquired from other organisms were qualitatively similar to those acquired with adult flies, suggesting that MARGO works with a variety of organisms.

We gave MARGO a graphical user interface (GUI) to make it accessible to users unfamiliar with MATLAB or programming in general (Fig 1D). We generally find that new users easily learn to use both the core work-flow and parameter customization. The typical setup time of a tracking experiment for trained users ranged between a few seconds (with saved parameter

profiles) to a few minutes (under novel imaging conditions). The utility of the GUI extends to customization of analysis, visualization, and input/output sources such as videos, cameras, displays, and COM devices. Descriptions and instructions for these use cases, including defining custom experiments via the API, are available in MARGO's documentation.

### Integrating hardware for closed-loop experiments

Real-time tracking allows the delivery of closed-loop stimuli that depend on the behavior of animals. MARGO offers native support for the hardware needed for closed-loop experiments including: cameras for real-time image acquisition, projectors/displays for visual stimuli, and serial COM devices for digital control of other peripheral electronics. COM devices include programmable microcontrollers (like Arduinos) that make it relatively simple to control a wide variety of devices. MARGO was designed to detect and communicate with such COM devices devices to integrate real-time feedback from sensors and coordinate closed-loop control of peripheral hardware.

We ran experiments with a custom circuit board to measure individual phototactic preference (the "LED Y-maze"). In this assay, individual flies explored symmetrical Y-shaped arenas with LEDs at the end of each arm (Fig 4A and 4B, S7 Video). For all arenas in parallel, real-time tracking detected which arm the fly was in at each frame. At the start of each trial, an LED was randomly turned on in one of the unoccupied arms. Once the fly walked into one of these two new arms, MARGO turned off all the LEDs in that arena. Immediately after these choice events, a new trial was initiated by randomly turning on an LED in one of the now unoccupied arms. This process repeated for each fly independently over two hours, and MARGO recorded which turns were toward a lit LED (positive phototaxis) and which were away (negative phototaxis) (Fig 4C). Tiling many such mazes on a single board yielded the experimental throughput for which MARGO is well-suited. Overall, we recorded choices from over 3,600 individuals, representing more than 830,000 choices in total.

To assess MARGO's capacity to reveal behavioral differences between genotypes, we tested a variety of wild type strains in the LED Y-maze. All strains exhibited a significant average positive phototactic bias (mean phototactic indices ranging from 0.55 to 0.80, p-values$<<10^{-6}$ by t-test). In contrast, blind flies (*Norp-A* mutants) and flies under identical circumstances but with unpowered LEDs, showed mean "preferences" indistinguishable from 0.5, consistent with random choices (Fig 4D). The wild type lines tested showed significant variation in population mean (one-way ANOVA; $F_{(6,1943)} = 118.2$, p$<<10^{-6}$) and population variability (one-way ANOVA on Levene-transformed data; $F_{(6,1943)} = 19.29$, p$<<10^{-6}$).

We collected LED Y-maze data from a single cohort of wild-type (Berlin-K, n = 144) flies over the first 8 days post-eclosion to profile phototaxis throughout development (Fig 4E). Flies displayed a significant average negative light bias (0.417, p$<<10^{-6}$) on the day of eclosion but transitioned to a positive light bias of 0.663 (p$<<10^{-6}$) by 7 days post-eclosion. This assay has structural similarities to an assay we previously used to measure locomotor handedness [28], the tendency of individuals to turn left or right when going through the center of the arena. In the LED Y-maze assay, locomotor left-right decisions were made in superposition with light-dark choices. Flies typically make hundreds of choices over the course of an experiment, giving us enough data to examine the turn bias of individuals in all four left-right/light-dark combinations. We divided trials into two groups based on whether the lit LED appeared to the right or left of the choice point. We found that the mean turn bias but not the mean phototactic bias differed between these two conditions (Fig 4F) [29]. Categorizing trials this way revealed that the rank order of both turn bias and phototactic bias are anti-

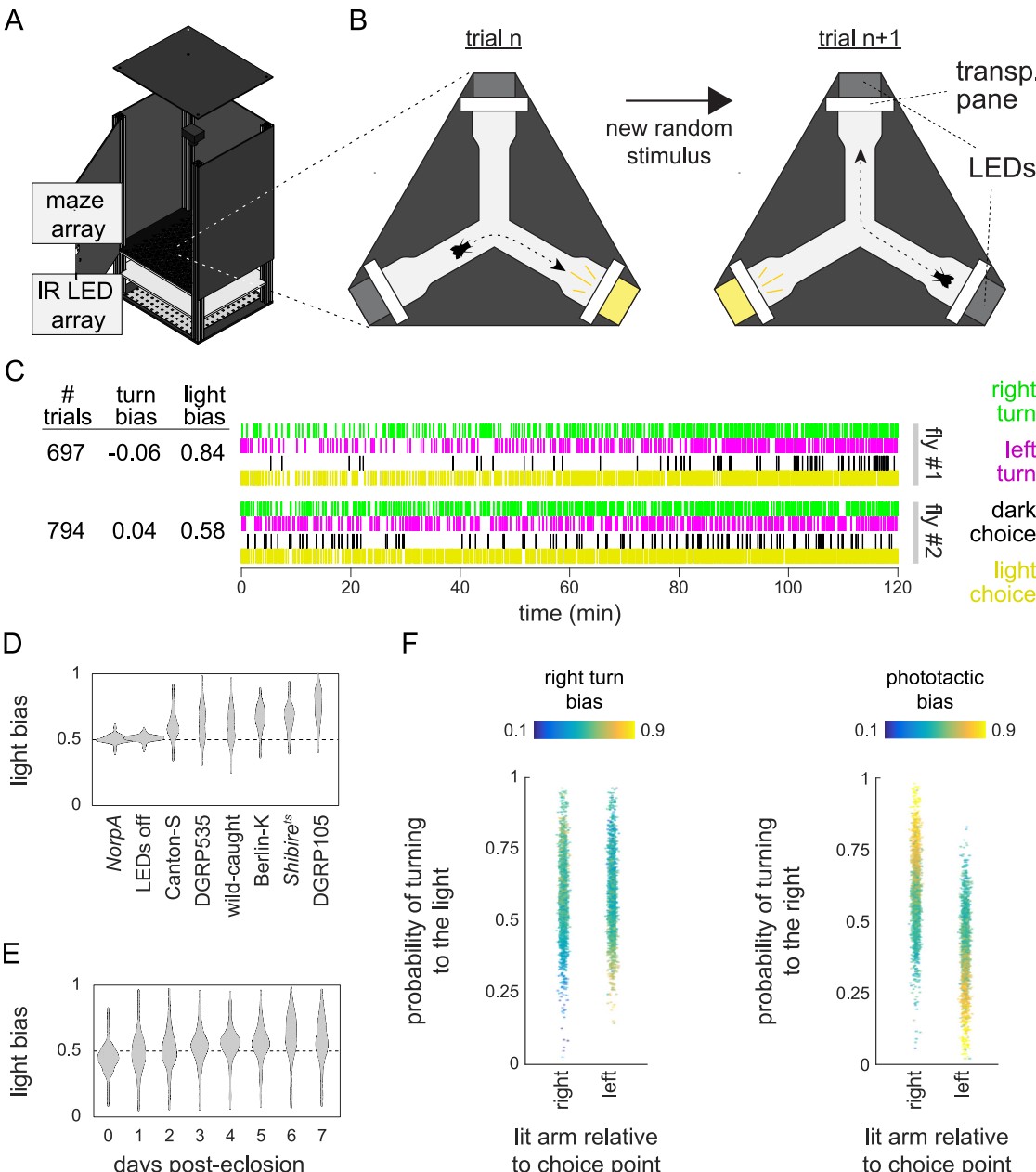

**Fig 4. High-throughput phototactic assay in Y-shaped arenas.** A) Schematic of the behavior box with an LED Y-maze array in place. B) Diagram of a single LED Y-maze and trial structure. New trials initiate by turning on (yellow) an LED in one of the two unoccupied maze arms. The trial ends when then animal turns into a new arm and the lit LED is turned off (gray). Each turn is scored for both handedness and phototactic preference. C) Raw turn data for two sample flies. Each individual trial consists of both a phototactic and handedness choice. Individual mean turn biases range from 0 (all left turns) to 1 (all right turns). Light biases range from 0 (all photopositive turns) to 1 (all photonegative turns). D) Comparison of individual average phototactic bias distributions for different wild-type fly lines. Blind flies (NorpA) and flies tested with all LEDs turned off (DGRP-105 dark) are included as negative controls. Horizontal dashed line indicates random bias at p = 0.5. E) Distribution of individual average phototactic biases for the same cohort of flies over the first 8 days post-eclosion. F) Individual mean phototactic and right turn biases calculated on all trials sub-divided by into trials where the lit arm of the maze was to the right or left of the choice point. Data points are colored by either the individual mean right turn bias (left panel) over all trials or the individual mean phototactic bias (right panel) over all trials. The rank orders of both turn bias and phototactic bias are anti-correlated (r = -0.38 and -0.63 respectively) between trials where right or left arm was lit.

correlated (r = -0.38 and r = -0.63 respectively) between the two conditions, suggesting that both individual phototactic bias and locomotor handedness bias affect each choice.

We adapted an optomotor paradigm [30] to a high-throughput configuration to test MARGO's ability to deliver a precise closed-loop stimulus with low latency. In this paradigm, an optomotor stimulus consisting of a high-contrast, rotating pinwheel, centered on a fly, is projected on the floor of the arena in which it is walking freely. On average, such optomotor stimuli evoke a turn in the direction of the rotation to stabilize the visual motion [31]. The center of the pinwheel follows the position of the fly as it moves around the arena so that the only apparent motion of the stimulus is around the fly. Thus, this stimulus is closed-loop with respect to each animal's position and open-loop with respect to its rotation velocity.

To implement this paradigm, we constructed a behavioral platform with a camera and an overhead mounted projector targeting an array of flat circular arenas (Fig 5A). To target a stimulus to a fly based on its coordinates in the tracking camera, MARGO had to learn the mapping of camera coordinates to projector coordinates. We added a feature to locate small dots displayed by the projector with the camera. From the position of these dots in camera coordinates, we constructed a registration mapping from the camera FOV to the projector display field. Using this mapping, we programmed MARGO to use the real-time positions of flies to project pinwheel stimuli independently to 48 freely moving individuals simultaneously (Fig 5B, S8 Video). To ensure faithful coordination between the tracking and stimulus, the tracking rate was matched to the refresh rate of the display at 60Hz (which is below the flies' flicker-fusion rate, meaning this stimulus produces beta movement apparent motion [32]; see Discussion).

While optimizing this assay, we observed that optomotor responses could be reliably elicited, provided individuals were already moving when the pinwheel was initiated. This is consistent with previous observations of optomotor responses depending on arousal state [33, 34]. We therefore configured MARGO to stimulate with the pinwheel each fly when: 1) it was moving 2), a minimum inter-trial interval had passed, and 3) it was a minimum distance away from the edge of the arena. The inter-trial interval helped prevent behavioral responses from adapting, and provided a baseline measurement period where no stimulus was present. Minimum distance to the edge ensured that the stimulus occupied a significant portion of the animal's field of view.

We characterized the optomotor behavior of wild type flies in a two hour experiment with two second pinwheel stimuli and a minimum inter-trial interval of 2s (Fig 5C). In total, over 300,000 trials were recorded from more than 1,800 flies, assayed in groups of up to 48 flies simultaneously. For each fly, we calculated an optomotor index [35] as the fraction (normalized to [-1,1]) of body angle change that occurred in the same direction as the stimulus rotation over the duration of the stimulus. On average, flies displayed reliable optomotor responses (mean index = 0.358, $p \ll 10^{-6}$) when stimulated with high-contrast pinwheels (Fig 5D). We observed significant individual variation in optomotor index (Fig 5E) as well as the number of trials each fly experienced, reflecting individual variation in the fraction of time walking.

To characterize the psychometric properties of this behavior, we randomly varied pinwheel contrast, angular velocity, and spatial frequency simultaneously on a trial-by-trial basis. Mean optomotor indices increased with pinwheel contrast, plateauing over much of the dynamic range of the projector, starting around 25% contrast (Fig 5F). Similarly, optomotor indices increased with both stimulus spatial frequency and angular speed, peaking at 0.18 cycles/degree and 360 degrees/s respectively (Fig 5G). The population mean optomotor index reversed at high combined values of spatial frequency and angular speed due to the apparent reversal of the stimulus at frequencies higher than the refresh rate of the projector.

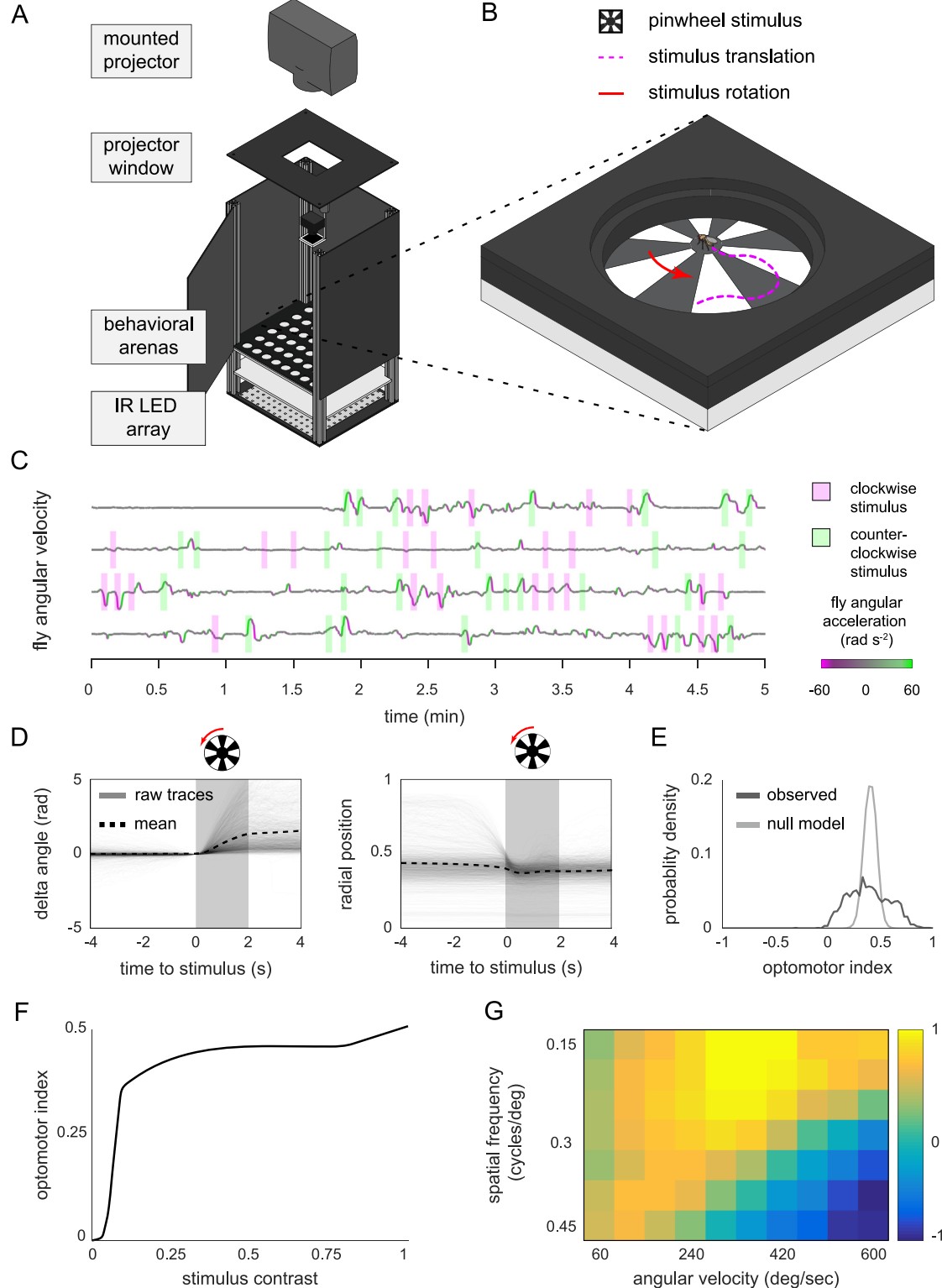

**Fig 5. High-throughput optomotor assay implementation in MARGO.** A) Schematic of the optomotor arenas and behavioral box. B) Diagram of a single arena and optomotor stimulus. Trials begin with a pinwheel stimulus, centered on the fly. For each trial, the rotational direction (red arrow) of the stimulus is randomized. As the animal moves, the pinwheel position is updated to stay centered on the fly. Trials end when the stimulus is removed after 2s. C) Four sample raw individual angular velocity time series. Flies typically respond to optomotor stimuli by turning in the direction of the rotation of the stimulus. Shaded rectangles indicate

the direction of pinwheel rotation, line color angular acceleration. D) Trial-triggered average optomotor response across all individuals. Change in body angle (left) is relative to body angle at stimulus onset. Sign indicates turns with (positive) or against (negative) the direction of stimulus rotation. E) Comparison of the observed distribution of individual average optomotor indices (n = 1,860) to the distribution expected under a null model in which all flies turn with identical statistics, generated by bootstrap resampling. F) Population average optomotor index as a function of stimulus contrast (0-1). Pinwheel contrast was randomly varied on a trial-by-trial basis. G) Average optomotor index as a function of stimulus spatial frequency and stimulus angular velocity.

## High-throughput optogenetic experiments

To test the versatility of MARGO, we used its API to implement high-throughput closed-loop optogenetic experiments using a digital projector to target individual flies expressing CsChrimson [36, 37] with flashing red light contingent on their behavior (Fig 6). We used a commericial Optoma S310e DLP projector which, when displaying red light ([255 0 0] RGB code), had a spectral range of 570 nm to 720 nm with a peak at 595 nm. Light stimulation frequency was set to the projector refresh rate (60Hz) and its intensity to the maximum, if not otherwise specified.

As a first experiment, we tracked the flies in a Y-Maze shaped like that in Fig 4A, but with no LEDs. Whenever a fly entered a designated arm, MARGO projected red light on it. Flies expressed CsChrimson in bitter-taste receptor neurons using the driver *Gr66a-GAL4*. MARGO recorded the fractional time spent in the lit arm (occupancy) and the number of entries into the lit arm (entries). We observed a modest increase in the aversive effects of optogenetic stimulation (reduced occupancy and entries) with light intensity (Fig 6A.1), whereas increasing stimulation frequency did not elicit any obvious change in aversion (Fig 6A.2). To test the robustness of the experiment to changes in the fictive conditioning stimulus, and to exclude the effects of visual cues, we expressed CsChrimson in heat sensitive neurons targeted by *Gr28bd+TrpA1-GAL4* in *norpA*$^{P24}$ blind flies. This experiment is conceptually analogous to spatial learning in the heat-box, where flies are trained to avoid one side of a dark, heatable chamber [38–45]. While blindness only marginally affected the time spent in the lit arm (the blind flies with Chrimson driven in heat-sensitive neurons still avoided occupying the lit arm at similar rates to seeing flies with Chrimson in bitter-sensitive neurons), the reduction in entries into the lit arm, observed in the seeing flies, was abolished (Fig 6A.3). These results suggest that vision is a key sensory modality informing the decision to enter an arm, but not for the decision of how much time to spend in an arm, once entered.

Analogous to a different heat-box experiment [46], optogenetic stimulation was made contingent on locomotor speed rather than position. In the same circular arenas as the optomotor experiments above (Fig 5A), the red light was switched on under two distinct conditions enforced in separate experimental blocks: 1) whenever the walking speed of the flies exceeded a threshold of 6.8 mm/s and 2) whenever the walking speed fell below that same threshold. The overall 64 minute experimental protocol consisted of 8 periods of 8 minutes each. The periods alternated between a baseline period, where the light was permanently switched off, and the two reinforcement periods where the light was contingent on either fast walking or slow walking/resting, respectively (Fig 6B.1 and 6B.2). As in the heat-box experiments, flies increased their walking speed when punished for walking too slowly. However, punishing fast walking failed to significantly decrease walking speed. Reminiscent of the induction of 'learned helplessness' in yoked control animals in the heat-box [46], flies trained with these conflicting schedules of punishment, significantly reduced their walking speed in the baseline periods without optogenetic stimulation, in comparison to control animals which did not express any CsChrimson (Fig 6B.3).

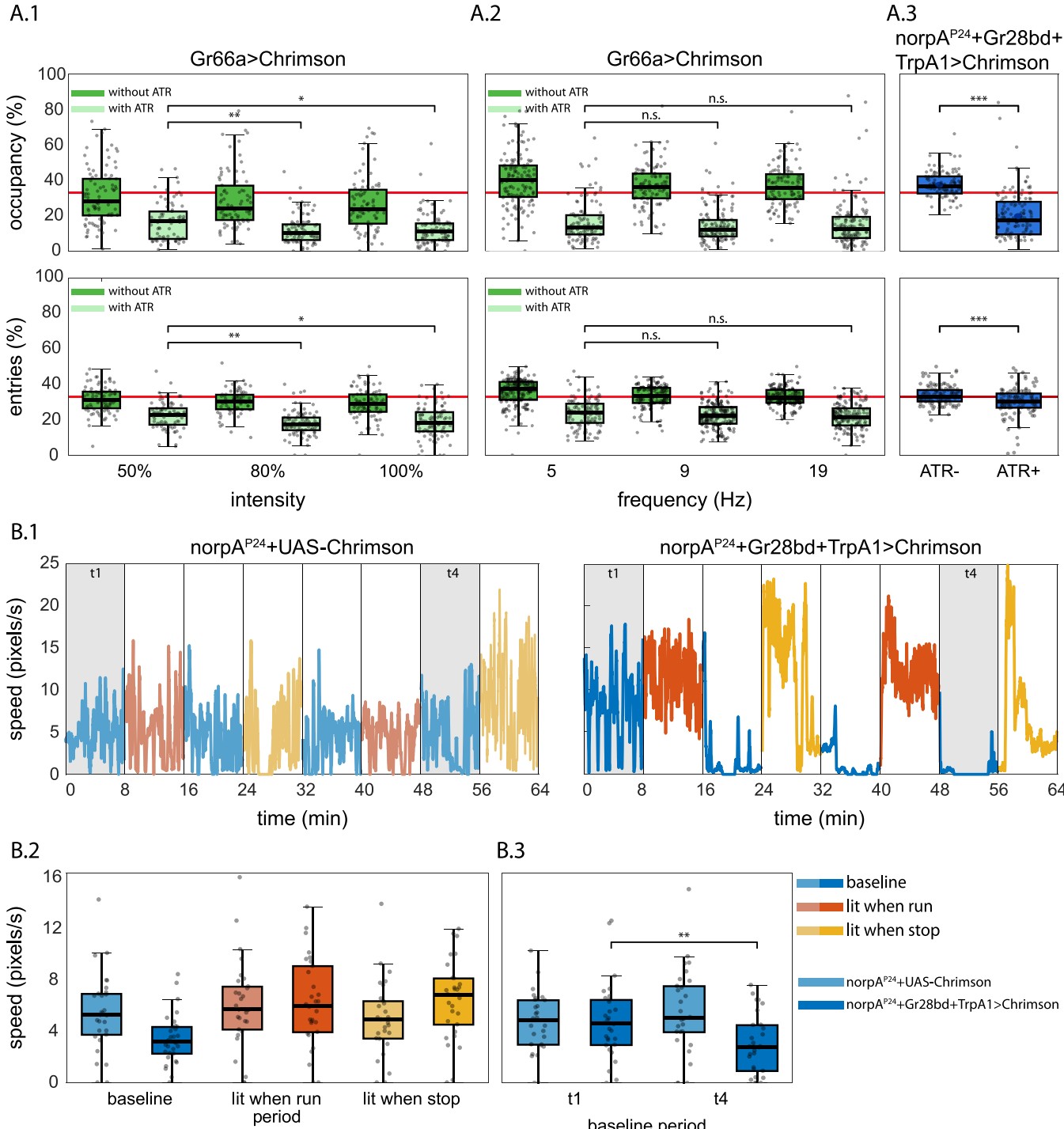

**Fig 6. Optogenetic closed-loop experiments with MARGO.** A.1) Top: Fraction of time spent in the arm of a Y-maze which was triggered to optogenetically stimulate flies expressing CsChrimson in bitter taste receptor neurons. Bottom: Portion of arm entries into the reinforced arm. Light green boxes are control flies not fed ATR; dark green experimental flies are fed ATR. Red line indicates chance rates. Individual points are flies. Even at the lowest intensity (50%), flies show a robust avoidance of the reinforced arm in a Y-Maze. Increasing light intensity (x-axis) further decreases (slightly) the lit arm occupancy time and the lit arm entries even further. Here and elsewhere *:p<0.05, **:p<0.01, ***:p<0.001. A.2) As in A.1, but varying the frequency of the optogenetic stimulation. Frequency had little effect on the occupancy or rate of entry into the reinforced arm. A.3) Blind *norpA^P24;Gr28bd+TrpA1>Chrimson* flies, expressing Chrimson in heat-sensitive neurons, also show decreased occupancy in the lit arm, whereas the fraction of entries into the lit arm appears unchanged compared to control flies not fed ATR. B.1) Example walking speed traces of an individual fly in circular arenas stimulated upon when above or below (depending on trial period) a speed threshold 4 px/s. Line color indicates which reinforcement paradigm was used in each period. Initial (t1) and final (t4)

baseline periods are highlighted (see B.3). Green line indicates the speed threshold. B.2) Walking speeds for all periods and all flies. *norpA$^{P24}$;Gr28bd+Trp A1>Chrimson* flies increase their walking speed specifically during periods when stimulation is contingent on slow walking or resting (lit when stop), compared to lit when running periods and controls without the optogenetic effector *norpA$^{P24}$;UAS-Chrimson*. B.3) Walking speed during the initial baseline period did not differ between experimental and control flies (t1). In contrast, after three reinforcement periods, walking speed in experimental flies was significantly lower than in control flies (t4). All flies in B were fed with all-trans-retinal.

## Discussion

We developed MARGO as a platform for a wide variety of behavioral paradigms and organisms, all at high throughput for large-scale experiments (like genetic screens, measuring individuality and characterizing psychometric response curves). MARGO's tracking algorithm, interface, and data footprint are lightweight, making it perform well in applications like real-time centroid tracking. Conversely, it is not made for harder computational tasks like maintaining the identity of multiple animals in the same compartment. But the ability to rapidly define ROIs, and track individuals in them, enables MARGO to easily coordinate low-latency, closed-loop stimulation for psychometric and optogenetic experiments. Furthermore, by packaging MARGO in a GUI and thoroughly documenting its usage and API, we hope to make it accessible both to new users with little programming experience and advanced users developing custom experimental paradigms.

When ROI boundaries are drawn along physical barriers, individual identities can be maintained indefinitely through ROI identity, thus removing the requirement for human supervision and intervention. We found that insisting on spatial segregation ultimately relaxes the computational requirements enough that thousands of individuals can be tracked in real time. In the future, real-time tracking that maintains individual identity without physical barriers may be possible, perhaps as an extension of current methodologies that exploit neural networks to track individuals offline [8, 13]. MARGO's interface assists in the automated definition of up to thousands of ROIs. An ROI-based architecture can also be used to distinguish groups rather than individual identities by separating groups into distinct arenas. This configuration therefore allows multiple groups, as well as individuals, to be tested in parallel.

Long-term automated behavioral measurement has great potential in the fields of sleep, circadian rhythms, pharmacology, and aging, among others. MARGO offers many features useful for activity measurement over long timescales, including rapid experimental setup, small data footprint, and built-in utilities for handling large data sets. For example, over a week we tracked the behavior of 960 flies simultaneously as they walked in the wells of custom 96-well plates (Fig 3). Such throughput can be applied to comparisons among individuals, genotypes, or treatment groups.

With built-in hardware support for cameras, displays, and peripheral electronics, MARGO enables open- and closed-loop stimulus-evoked ethology on a large scale. Built-in features supporting projector displays, like camera-projector registration, facilitate a wide variety of visual and optogenetic experiments (Figs 5 and 6). Native detection and communication with serial COM devices further extends these capabilities by providing a generic interface for a wide variety of peripheral devices, such as the LED controllers we used for the LED Y-maze (Fig 4). Taken together, MARGO is a multi-purpose platform for coordinating hardware inputs and outputs for high-throughput ethology.

Between our two closed-loop visual stimulation experiments (LED Y-maze and optomotor assay), we screened nearly 5,000 animals over hundreds of thousands of trials, allowing the precise characterization of both individual- and population-level behavior. With the experiments themselves representing less than a week of testing, these platforms could be used for large behavioral screens of hundreds of strains. In the LED Y-maze, we showed that

individuals displayed idiosyncratic biases in both phototactic preference and locomotor hand-edness simultaneously, as observed previously in separate assays [28, 47]. The wild-type fly lines we screened displayed population level differences in both mean preference and variabil-ity in phototactic bias [29]. Furthermore, the mean of one strain (Berlin-K) shifted from nega-tive to positive over the first week post-eclosion, as was reported previously [48]. Interestingly, we observed that flies with a high right-turn probability were more likely to turn toward the light when it was to the right of the choice point and that the opposite was true of flies with a high left-turn probability. We observed a similar but stronger effect of phototactic bias on locomotor handedness (e.g. flies with a high phototactic bias were more likely to turn toward the right when the light was on the right). Together these results demonstrate measurable effects of phototactic bias and handedness in a task that probes both simultaneously. Thus, we found that both individual light and handedness biases influence light/turn behavior on a choice-by-choice basis. As responses to light are ethologically relevant [49], the interplay of individual behavioral biases may have fitness consequences for wild flies.

In the optomotor experiment, we demonstrated that, using closed-loop stimuli delivered from a projector, MARGO can quantify individual optomotor responses of dozens of flies simultaneously. Consistent with previous findings [33, 34], we saw that stationary flies did not exhibit strong optomotor responses, consistent with the idea that this reflexive behavior may be state-dependent [50–54]. While all animals tested exhibited the optomotor response to some degree, we observed a broad distribution of individual optomotor indices, suggesting that individuals respond idiosyncratically to the same stimulus, as has been found previously in other spontaneous and stimulus-evoked behaviors [17, 28, 47, 49, 55]. We suspect that the success of this assay may be partially due to tightly centering the pinwheel centered on the ani-mal as it moves, which is possible because of MARGO's low latency.

Our optogenetic experiments provide a proof of principle that high-throughput closed-loop manipulation of neural activity is feasible (Fig 6). Using different driver lines to activate neu-rons under both spatial (Fig 6A) and locomotor (Fig 6B) contingencies, optogenetic stimula-tion reliably altered fly behavior in the expected directions. These experiments also revealed that flies use visual elements of the projector rig to orient when the stimulus was nominally off, and that optogenetic punishment can induce learning effects outlasting the stimulation itself. These results also remind us about a general limitation of studying freely moving animals: the large number of degrees of freedom that such behavior enables can make it difficult to causally relate biological manipulations to specific mechanisms. For instance, without prior knowledge of the function of the optogenetically targeted neurons, it would not have been immediately clear if our manipulation affected reinforcing neurons or neurons involved in motor control, which could also lead to altered occupancy of the lit arm in the Y-Maze. Likewise, a screen for neurons that are required for non-random entry into optogenetically-reinforced arms of the Y-maze would yield blind flies, as the flies in our assay apparently use visual cues to identify which arms are reinforced before entering them.

Behavioral experiments are frequently more complex than tracking objects in a dish. Such experiments could require complex arena geometries, data streams from external sensors, con-trol of peripheral hardware, and access to measurements of behavior in real time. Bonsai is an open-source, visual programming framework for combining input and output streams of mul-tiple devices such as cameras, microcontrollers, and other peripheral devices and defining experimental architecture from beginning to end. MARGO can manage these same features, making both programs well suited to implementing new behavioral paradigms. As a point of contrast, MARGO makes assumptions about the core workflow of the experiment and there-fore only offers customization at a particular node in the workflow (between tracking and data output on each frame). We designed MARGO around a workflow what we think is likely to be

**Table 1. Comparison of open-source animal tracking packages.** Trackers as falling into two rough categories: 1) real-time trackers capable of very high throughput and potential hardware integration, and 2) offline trackers capable of tracking body parts and/or maintaining individual identities without spatial segregation.

| | MARGO | Ethoscopes | BioTracker | Bonsai | ToxTrac | flyTracker | Ctrax | idTracker | Tracktor |
|---|---|---|---|---|---|---|---|---|---|
| Real-time | real-time or offline | real-time | real-time or offline | real-time or offline | offline | offline | offline | offline | offline |
| Peripheral hardware integration | yes | yes | camera only | Yes | no | no | no | no | no |
| Supports experimental models | yes | yes | no | depends | no | no | no | no | no |
| Custom hardware required | none | ethoscope | none | none | none | none | none | none | none |
| Track multiple animals per ROI | yes | yes | yes | depends | yes | yes | yes | yes | yes |
| Resolves identity through collisions | no | no | no | depends | yes | yes | yes | yes | no |
| Limb, head, midline, wing tracking | no | no | no | depends | no | yes | yes | yes | no |
| GUI | yes | yes | yes | depends | yes | yes | yes | yes | no |
| ROI definition | automatic or grid | grid or manual | depends | automatic or manual | automatic | grid or manual | single ROI | manual drawing | manual definition |
| Notes | fast tracking with hardware control | low-cost, easily-scalable hardware module | allows user defined tracking algorithms | visual programming to combine many data streams | simple setup and UI (MS Windows only) | uses feature detectors to track body parts | widely used, has behavioral analysis toolbox | maintains many individual IDs | well-suited to non-static background |

useful to many users, and can be tweaked to meet specific experimental needs, hoping that this structure will simplify implementation of custom experiments. Specifically, MARGO can automatically generate templates for new experiments with custom inputs and outputs within the GUI. We have also included a tutorial for defining custom experiments in MARGO's documentation. In practice, we find that new experiments can typically be defined in one or two custom functions, given familiarity with the API.

Animal tracking platforms are evolving to meet the diverse needs of the ethology, neuroscience and behavioral genetics communities. See Table 1 for a comparison of features of several contemporary tracking programs. Trackers can be broadly described as falling into one of two categories: 1) real-time trackers [1, 12, 19, 20, 22, 24] with potential for high throughput and hardware control and 2) offline trackers [5–8, 14, 15] with the potential to maintain individual identities (without using spatial segregation) and/or track body parts. Hardware integration is a natural extension of real-time trackers since many stimulus paradigms are contingent on behavior. While trackers in the second category are currently unsuitable for real-time applications, they offer the notable benefits of being able to study fine-scale postural and social behaviors. The ability to record video in parallel with tracking and peripheral hardware control means MARGO can be used upstream of offline trackers, making it possible to analyze social dynamics or postural features in response to closed-loop stimuli. Among this array of options, MARGO is optimized for the throughput characteristic of *Drosophila* and other genetic model organisms like *C. elegans*. MARGO has the flexibility to accommodate the experimental diversity of techniques in neuroethology. Thus, we envision MARGO's niche as a versatile platform

for experiments operating at high throughput to measure individual behavior and deliver closed-loop sensory and optogenetic stimuli.

## Methods

### Repositories

MARGO's code is available in the MARGO repository on github. All behavioral data is available on Zenodo. Instrument schematics are available on github at: de Bivort Lab schematics repository.

### Software

The MARGO GUI, tracking algorithm, and all analysis software were written in MATLAB (The Mathworks, Inc, Natick, MA). Detailed descriptions of the functions and use of the MARGO GUI, ROI detection, background referencing, tracking implementation, noise correction, and data output are available in MARGO's documentation. Optomotor stimuli were crafted and displayed using the Psychtoolbox-3 for MATLAB. Software for control of all custom electronic hardware was written in C using Arduino libraries.

### Organism genotypes and rearing

Unless otherwise specified, the genotype of all fruit flies tested was a strain of Berlin-K that we inbred for 13 generations prior to these experiments. Gr66a-G4 (from the G. Turner lab), *norpA$^{P24}$* (from the M. Heisenberg lab), TrpA1-G4 (FlyBase ID: 27593), Gr28bd-G4 (FlyBase ID: 57620), UAS-20xCsChrimson (FlyBase ID: 55135) were the lines used in the optogenetic experiments. Tracking experiments were performed with mixed sex flies 3-5 days post-eclosion unless otherwise noted. Flies were raised on standard conrmeal/dextrose formula media (Harvard Fly Core Facility) under 12 h/12 h light and dark cycle in an incubator at 25˚C and 40% humidity. Animals were imaged and singly-housed on food in modified 96 well plates (Fly Plates, FlySorter LLC) for all multi-day tracking experiments. *C. Elegans* were housed in a custom platform on agarose media and were composed of multiple strains as described in the WorMotel publication [56]. *Drosophila* larvae CantonS on 2% agarose media mixed with fructose in a gradient (0-300mM) along one axis. Larval zebrafish were *HC:GCaMP6s*.

### Behavioral acquisition

All real-time tracking images were captured with USB or GigE cameras from Point Grey (Firefly MV 13SC2 and BFLY-PGE-12A2M-CS). Images for the minimal hardware setup were acquired with an iPhone 5s 1.3MP camera. Cameras used in real-time experiments were fitted with a long-pass 87 Kodak Wratten infrared filter with a cutoff frequency of 750nm and illuminated with infrared LEDs centered at 940nm (Knema LLC, Shreveport, LA). Acquisition rates varied by experiment between 5.0 fps for LED Y-maze and 60.0 fps for optomotor response. Flies were imaged at spatial resolutions ranging between 1-4 pixels per mm and we generally found tracking to be stable at 10 pixels per animal and above. Offline video tracking was performed on 1000x compressed AVI video files unless otherwise specified. Tracking and imaging was conducted in Windows 10 on computers with CPUs ranging from intel i3 3.1GHz to intel i7 4.0GHz.

### Behavioral instruments

Unless otherwise specified, all tracking was conducted in custom imaging boxes constructed with laser-cut acrylic and aluminum rails. Schematics of custom behavioral arenas and

behavioral boxes were designed in AutoCAD. Arena parts were laser-cut from black and clear acrylic and joined with Plastruct plastic weld. Schematics for behavioral boxes can be found on the de Bivort Lab schematics repository. Illumination was provided by dual-channel white and infrared LED array panels mounted at the base (Part# BK3301, Knema LLC, Shreveport, LA). Adjacent pairs of white and infrared LEDs were arrayed in a 14x14 grid spaced 2.2cm apart. White and infrared LEDs were wired for independent control by MOSFET transistors and a Teensy 3.2 microcontroller. Two sand-blasted clear acrylic diffusers were placed in between the illuminator and the behavioral arena for smooth backlighting. Additional tracking was performed in standard 48 multi-well culture plates and individual fly storage units (FlyPlates) from FlySorter LLC. Additional details on the behavioral platforms used here are available in the MARGO documentation.

## Experimental procedures

Tracking experiments were conducted between 10AM and 6PM. We saw no time-of-day significant effects on individual behavioral measures from the optomotor and LED Y-maze assays. Flies were anesthetized either on ice or $CO_2$ and manually loaded into behavioral arenas with an aspirator. Behavioral modules were loaded into tracking boxes and allowed a minimum post-anesthesia recovery period of 20 minutes before tracking. Unless otherwise specified, animals were tracked for 2 hours in an environmental room at 23 C and 40% humidity. Following tracking, flies were returned to individual storage plates where they were housed for further experiments as needed. For the optogenetic experiments, flies were tested for 20 min in the Y-mazes or 64 min in the circular arenas.

## Data and statistics

Unless noted, all reported error bars are 95% confidence intervals computed by bootstrap resampling. Data processing and calculation of behavioral metrics was conducted automatically by MARGO either in real time, or after experiments. 1000 bootstrap replicates were averaged to estimate null distributions and confidence intervals. Reported p-values for phototaxis, optogenetic closed-loop experiments and optomotor behavior were unadjusted for multiple comparisons and were calculated via two-tailed t-tests. Critical values were adjusted for multiple comparisons via Bonferroni correction.

## Supporting information

**S1 Video.** https://youtu.be/fyG31BAYHE0. Video from 960 fly experiment with MARGO traces overlaid. Inactive flies are unmarked.
(MP4)

**S2 Video.** https://youtu.be/0aFny65wCnM. MARGO tracking of flies in a simple imaging configuration made from a cardboard box and a sheet of paper. Video was captured with a 1.3MP iPhone camera and tracked offline.
(MP4)

**S3 Video.** https://youtu.be/M8imxRP92k4. MARGO tracking of 48 larval zebrafish in a multi-well culture plate (2x speed).
(MP4)

**S4 Video.** https://youtu.be/kuTM71lHALc. MARGO tracking of *C. Elegans* in WorMotel, a custom 2400 well platform for studying aging.
(MP4)

**S5 Video.** https://youtu.be/sxQMXHJoG24. MARGO tracking of 38 fruit fly larvae in a single ROI in response to a fructose gradient. Individual identities are not maintained through collisions.
(MP4)

**S6 Video.** https://youtu.be/FVIXQSdiWx0. MARGO tracking of a time-lapsed video of a Bumblebee colony in an artificial nestbox. Due to the low temporal resolution, individual identities are not maintained at all.
(MP4)

**S7 Video.** https://youtu.be/PqPJA6hsabE. Summary video of the image processing, object tracking, and closed-loop control in the LED Y-Maze assay.
(M4V)

**S8 Video.** https://youtu.be/uxgswI8jEWY. Summary video of the image processing, object tracking, and closed-loop control in the Optomotor assay.
(MP4)

## Acknowledgments

We thank Ed Soucy and Joel Greenwood for help troubleshooting and fabricating the LED Y-maze board; Simon Forsberg, Jamilla Akhund-Zade, and Carolyn Elya for providing crucial beta testing and feedback during MARGO development; Vladislav Susoy and Robert Johnson for donating worms, and fish to test MARGO tracking in other species; Jessleen Kanwal, James Crall and Matthew Churgin for donating movies of larvae, bees and worms for tracking. ZW was supported by an NSF Graduate Research Fellowship (DGE-1144152). CR was supported by iPUR and Fulbright fellowships. BdB was supported by the NSF (IOS-1557913), the Alfred P. Sloan Foundation, The Klingenstein-Simons Fellowship, and the Smith Family Foundation.

## Author Contributions

**Conceptualization:** Zach Werkhoven, Christian Rohrsen, Björn Brembs, Benjamin de Bivort.

**Data curation:** Zach Werkhoven.

**Formal analysis:** Zach Werkhoven, Benjamin de Bivort.

**Funding acquisition:** Björn Brembs, Benjamin de Bivort.

**Investigation:** Zach Werkhoven, Christian Rohrsen, Chuan Qin, Benjamin de Bivort.

**Methodology:** Zach Werkhoven, Christian Rohrsen, Chuan Qin, Björn Brembs, Benjamin de Bivort.

**Project administration:** Benjamin de Bivort.

**Resources:** Zach Werkhoven, Christian Rohrsen, Chuan Qin, Benjamin de Bivort.

**Software:** Zach Werkhoven, Chuan Qin, Benjamin de Bivort.

**Supervision:** Zach Werkhoven, Benjamin de Bivort.

**Validation:** Zach Werkhoven, Christian Rohrsen, Benjamin de Bivort.

**Visualization:** Zach Werkhoven, Christian Rohrsen, Benjamin de Bivort.

**Writing – original draft:** Zach Werkhoven, Christian Rohrsen, Björn Brembs, Benjamin de Bivort.

**Writing – review & editing:** Zach Werkhoven, Christian Rohrsen, Björn Brembs.

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
