## [Editor Report · Decision Letter 0]

21 Jun 2019

PONE-D-19-17145

MARGO (Massively Automated Real-time GUI for Object-tracking), a platform for high-throughput ethology

PLOS ONE

Dear Dr. de Bivort,

Thank you for submitting your manuscript to PLOS ONE. I have reviewed the comments this manuscript received during its previous submission on the sister journal and I am happy to proceed using the feedback from those reviewers together with my own judgement. Reviewer #3, whose comments are here attached once more for your convenience, raised some questions aimed at clarifying some technical aspects of the work and we invite you to submit a revised version of the manuscript that addresses the points raised during the review process.

We would appreciate receiving your revised manuscript by Aug 05 2019 11:59PM. To enhance the reproducibility of your results, we recommend that if applicable you deposit your laboratory protocols in protocols.io, where a protocol can be assigned its own identifier (DOI) such that it can be cited independently in the future. For instructions see: http://journals.plos.org/plosone/s/submission-guidelines#loc-laboratory-protocols

We look forward to receiving your revised manuscript.

Kind regards,

Giorgio F Gilestro, PhD

Academic Editor

PLOS ONE

Journal Requirements:

I have read the journal's policy and the authors of this manuscript have the following competing interests: BdB is a scientific advisor at FlySorter, LLC

We note that you received funding from a commercial source: [Name of Company]

Additional Editor Comments:

Reviewer #3: (inherited from previous submission)

The authors propose a system to track many animals independently. It is particularly tuned for high throughput of independent animals.

Major

It is not clear to me that Bonsai cannot do what MARGO does. Could you compare?

I could not find some parts of the algorithm (how it is decided that a blob is noise or animal, how you go from centroids to trajectories)

Minor

Authors seem to confuse idTracker with idtrackrer.ai. It is the second one that uses training, while you claim is the first one. Also idtracker.ai seem to be missing in ‘while others measure the collective

activity of groups without maintaining identities or rely on physical segregation of animals to ensure trajectories never Collide’

Unclear what ‘sample imaging statistics of clean tracking’ means

‘Each background image is computed as the mean or median image from a rolling stack of background sample images.’ How do you choose between mean or median?

Does the tracking start independently for every ROI? How long does it take to construct the BKG image? If an animal never moves, is there a way to extract this information or it should be inferred by the number of tracks obtained?

Typo: ‘pvalues¡¡

10−6 by t-test).’ And similar ones. pvalues¡¡ 10−6 by t-test).

I think the Abstract needs to be more specific. For example: ‘tracking’ is for animals independetly, for example in different wells. Maybe also clarify it is for MATLAB, as this is very informative for users. In main text I learn much better but MARGO is about:

1) fast and accurate individual tracking that

could be scaled to very large numbers of individuals or

experimental groups over very long timescales, 2) flexibility

in the user interface that would permit a diversity of

organisms, tracking modes, experimental paradigms, and

behavioral arenas, 3) integration of peripheral hardware to

enable closed-loop sensory and optogenetic stimuli, and 4)

a user-friendly interface and data output format.
---

## [Author Response · Author response to Decision Letter 0]

7 Oct 2019

[This is a copy of the response text I uploaded as a component of the revision]

Please find below a the full set of comments from the previous reviewer and editor (in blue) and our replies (in black). These were helpful comments and we have implemented changes that have improved the manuscript. 

[From the editor:]

> Thank you for submitting your manuscript to PLOS ONE. I

> have reviewed the comments this manuscript received during

> its previous submission on the sister journal and I am

> happy to proceed using the feedback from those reviewers

> together with my own judgement. Reviewer #3, whose

> comments are here attached once more for your convenience,

> raised some questions aimed at clarifying some technical

> aspects of the work and we invite you to submit a revised

> version of the manuscript that addresses the points raised during the review process.

We thank the editor and reviewer for their constructive feedback, and believe we have made appropriate revisions to the manuscript. 

[From Reviewer 3]

The authors propose a system to track many animals independently. It is particularly tuned for high throughput of independent animals.

Major Comments

It is not clear to me that Bonsai cannot do what MARGO does. Could you compare?

The reviewer rightly points out that Bonsai is similar to MARGO in the sense that it features support for many hardware devices and a flexible interface for defining custom experiments. We did not originally include Bonsai in our list of trackers because Bonsai is closer to a programming environment than a tracker. We are not experts in Bonsai and cannot definitively say what Bonsai can do, but we believe, given the flexibility of Bonsai and ReactiveX, that most if not all of MARGO’s functionality could be written and implemented in Bonsai. The flexibility offered by Bonsai extends well beyond that of MARGO. We see MARGO as a specific implementation of a tracking interface that includes many features useful for high-throughput and real-time tracking. MARGO does offer some flexibility for defining new experimental regimes within the same overarching schema of parallel tracking and/or stimulus targeting of hundreds or thousands of ROIs. We believe that some users may find it simpler and more efficient to run experiments similar to those pre-packaged with MARGO, particularly users who are unfamiliar with Bonsai’s API and ReactiveX.

We have now added Bonsai to our table of trackers, noting its generality as a device programming environment. We have also now included it in the discussion. 

I could not find some parts of the algorithm (how it is decided that a blob is noise or animal, how you go from centroids to trajectories)

We thank the reviewer for pointing out this discrepancy and have added clarifying text. Briefly, in each frame, candidate blobs are identified as the blobs that are both 1) between minimum and maximum size threshold and 2) located within the bounds of an ROI. Candidate blobs are subsequently assigned to ROIs by spatial location. Within each ROI, candidate blobs are matched to centroid traces by minimizing the total frame-to-frame changes in position within each ROI. If the number of candidates exceeds the number of traces in a given ROI, only the candidates closest to the last centroid positions of the traces are assigned. If the number of traces exceeds the number of candidates, the candidates are assigned to the closest traces and any remaining traces are assigned no position (i.e. NaN) for that frame.

Minor Comments

Authors seem to confuse idTracker with idtrackrer.ai. It is the second one that uses training, while you claim is the first one. 

We have changed the text and citation to correctly cite idTracker.ai.

Also idtracker.ai seem to be missing in ‘while others measure the collective activity of groups without maintaining identities or rely on physical segregation of animals to ensure trajectories never Collide’

We believe that idTracker.ai does not belong on this list, since it is capable of maintaining individual identities without physical separation.

Unclear what ‘sample imaging statistics of clean tracking’ means.

We have re-worded this sentence clarify that this step of the MARGO workflow refers to sampling the number of foreground pixels under clean imaging conditions to construct a distribution and compute statistics on that distribution.

‘Each background image is computed as the mean or median image from a rolling stack of background sample images.’ How do you choose between mean or median?

The user can choose within the tracking options whether the background image is computed as the mean or median of the sample images. By default, MARGO computes the background image as the median of the sample images since it can compute an accurate representation of how the background should look without the animal with as few as three images (provided that the animal moves between samples). Mean referencing often requires more sampling to remove ghost-like images of the animal from the background. We have reworded the sentence to help clarify that point.

Does the tracking start independently for every ROI? 

Tracking starts for all ROIs simultaneously once the setup steps are completed (ie. ROI definition, background image, and . 

How long does it take to construct the BKG image?

Typically we find that it takes between 3-30 seconds to establish the background image for most animals. Importantly, background image updating occurs continuously throughout tracking; so it is not essential that animals move prior to the start of tracking. We have now added this information to the text.

If an animal never moves, is there a way to extract this information or it should be inferred by the number of tracks obtained?

The number of traces output by MARGO is specified by the user prior (default) or automatically estimated (if this option is selected by the user) prior to tracking. In either case, the total number of traces is fixed before tracking and are organized by ROIs (the number of traces in each ROI can be a different number for each ROI). Animals that never move and are therefore averaged into the background, will receive no assigned position in each frame (i.e. NaN). Thus there will be a trace for each animal and it is possible to identify which animals are inactive by the total distance traveled for each trace. We’ve clarified the above points in the text.

I think the Abstract needs to be more specific. For example: ‘tracking’ is for animals independently, for example in different wells. Maybe also clarify it is for MATLAB, as this is very informative for users. In main text I learn much better but MARGO is about:

1. fast and accurate individual tracking that could be scaled to very large numbers of individuals or experimental groups over very long timescales

2. flexibility in the user interface that would permit a diversity of organisms, tracking modes, experimental paradigms, and behavioral arenas

3. integration of peripheral hardware to enable closed-loop sensory and optogenetic stimuli

4. a user-friendly interface and data output format.

We have attempted to make the abstract more specific, and adopted the reviewer’s suggestion to indicate that it is a MATLAB package. Since MARGO does track multiple animals in the same well/arena (though without maintaining identity), we didn’t want to imply that it exclusively tracks one animal per well, but we changed the language to emphasize that this is the typical configuration. We also modified the abstract to better highlight the specific features listed above.

---

## [Editor Report · Decision Letter 1]

9 Oct 2019

MARGO (Massively Automated Real-time GUI for Object-tracking), a platform for high-throughput ethology

PONE-D-19-17145R1

Dear Dr. de Bivort,

We are pleased to inform you that your manuscript has been judged scientifically suitable for publication and will be formally accepted for publication once it complies with all outstanding technical requirements.

With kind regards,

Giorgio F Gilestro, PhD

Academic Editor

PLOS ONE
---

## [Editor Report · Acceptance letter]

6 Nov 2019

PONE-D-19-17145R1 

MARGO (Massively Automated Real-time GUI for Object-tracking), a platform for high-throughput ethology 

Dear Dr. de Bivort:

I am pleased to inform you that your manuscript has been deemed suitable for publication in PLOS ONE. Congratulations! Your manuscript is now with our production department. 

With kind regards,

on behalf of

Dr. Giorgio F Gilestro 

Academic Editor

PLOS ONE